# Conceptual frameworks for understanding the acceptability and feasibility of the minimally invasive autopsy to determine cause of death: Findings from the CADMIA Study in western Kenya

Kelvin Oruko[1,2]*, Maria Maixenchs[3,4], Penelope Phillips-Howard[5], Maureen Ondire[1], Clarah Akelo[1], Ariadna Sanz[3], Jaume Ordi[3,6], Clara Menéndez[3,4,7], Quique Bassat[3,4,8,9], Frank O. Odhiambo[1‡], Khatia Munguambe[4,10‡]

1 Kenya Medical Research Institute, Centre for Global Health Research, Kisumu, Kenya, 2 Kenya Medical Training College, Nairobi, Kenya, 3 ISGlobal, Hospital Clínic – Universitat de Barcelona, Barcelona, Spain, 4 Centro de Investigação em Saúde de Manhiça (CISM), Maputo, Mozambique, 5 Liverpool School of Tropical Medicine, Liverpool, United Kingdom, 6 Department of Pathology, Hospital Clinic – Universitat de Barcelona, Barcelona, Spain, 7 Consorcio de Investigación Biomédica en Red de Epidemiología y Salud Pública (CIBERESP), Madrid, Spain, 8 Catalan Institution for Research and Advanced Studies (ICREA), Barcelona, Spain, 9 Pediatric Infectious Diseases Unit, Pediatrics Department, Hospital Sant Joan de Déu (University of Barcelona), Barcelona, Spain, 10 Faculty of Medicine, Universidade Eduardo Mondlane (UEM), Maputo, Mozambique

‡ These authors are joint senior authors on this work.
* kelvinoov@gmail.com, koruko@kmtc.ac.ke

**Data Availability Statement:** Data cannot be shared publicly because of the institutional

## Abstract

Establishing the cause of death (CoD) is critical to better understanding health and prioritizing health investments, however the use of full post-mortem examination is rare in most low and middle-income counties for multiple reasons. The use of minimally invasive autopsy (MIA) approaches, such as needle biopsies, presents an alternate means to assess CoD. In order to understand the feasibility and acceptability of MIA among communities in western Kenya, we conducted focus groups and in-depth interviews with next-of-kin of recently deceased persons, community leaders and health care workers in Siaya and Kisumu counties. Results suggest two conceptual framework can be drawn, one with facilitating factors for acceptance of MIA due to the ability to satisfy immediate needs related to interest in learning CoD or protecting social status and honoring the deceased), and one framework covering barriers to acceptance of MIA, for reasons relating to the failure to serve an existing need, and/or the exacerbation of an already difficult time.

## Introduction

Identification of the cause of death (CoD) is challenging in low- and middle- income countries (LMIC). Bereaved families receive little information due to limited access to medical services, whereas at the societal level death registries are usually incomplete and pathology resources are often extremely limited or nonexistent. Such challenges can also be encountered in Kenya, thus

protection of participants, as the participants have not provided informed consent for their data to be stored in a public repository. Data are available from KEMRI/CDC Research and Public Health Collaboration data base. Data can be requested though the e-mail address kegapads@cdc.gov (Project Local Social Science Investigator of the CaDMIA study – Kenya).

**Funding:** The CaDMIA research project (Validation of the minimally invasive autopsy tool for cause of death investigation in developing countries) is funded by the Bill & Melinda Gates Foundation (Global Health grant numbers OPP1067522 and OPP1128001) and by the Spanish Instituto de Salud Carlos III (FIS, PI12/00757; CM, Acciones CIBER). ISGlobal is included in the CERCA Programme /Generalitat de Catalunya CISM is supported by the Government of Mozambique and the Spanish Agency for International Development (AECID). QB has a fellowship from the program Miguel Servet of the ISCIII (Plan Nacional de I+D+I 2008-2011, grant number: CP11/00269). The funders had no role in study design, data collection and analysis, decision to publish, or preparation of the manuscript.

**Competing interests:** The authors have declared that no competing interests exist.

hindering the accurate estimation of cause-specific mortality rates. CoD investigation in some LMIC settings employs verbal autopsy (VA), which is valuable but has well recognized limitations [1, 2]. Studies have shown that clinical diagnosis is less accurate compared to complete autopsy [3–6]. Additionally, the frequent clinico-pathological discrepancies identified in studies conducted in the developing world (e.g., autopsy studies of maternal deaths in Mozambique [7], and paediatric deaths in Malawi [8]), emphasize the limitations of using hospital-based clinical data, and the need for more precise biological methods to improve ascertainment of CoD. Improved data on the causes of mortality in LMIC settings are needed to inform public health planning and targeting of resources. While complete diagnostic autopsy (CDA) is considered the "gold standard" method, [9], routine use of this approach is difficult in rural (and even many urban) areas of LMIC due to lack of available expertise [10], limited suitable facilities, complex logistics, costs, and lack of familiarity with, and at times unacceptability, of the procedure [11].

Less invasive approaches as an alternative to CDA may be more feasible as well as more acceptable in LMIC [12]. One such approach is the minimally invasive autopsy (MIA), also referred to as minimally invasive tissue sampling (MITS), which uses targeted small diagnostic biopsies of key organs and fluids for histological, microbiological and molecular analyses [9]. Results suggest that in many cases MIA can produce reliable and comparable results to the CDA, especially when communicable diseases account for a significant proportion of deaths [13–16].

Understanding the cultural and religious beliefs and practices related to death in LMIC settings is critical before implementation of MIA. Of particular concern is the perspective of next-of-kin, who must be willing to allow MIA to be performed. Multiple factors including fear of body mutilation, concern over whether the deceased would have been willing to consent, cultural or religious norms, age of the deceased, and delays in funeral procedures are thought to be important factors affecting acceptance of post-mortem procedures [14–16]. Medical professionals' awareness about the procedure, relevance and benefits, as well as their skills and attitudes towards approaching a family grieving the loss of a member could also influence families' and communities´ acceptance of MIA [17–20]. The degree to which these factors apply to the use of MIA, particularly in our setting in East Africa, has been less comprehensively documented.

In preparation for a multi-centre mortality surveillance using MIA, ethnographic feasibility and acceptability studies were conducted to understand attitudes, beliefs and practices related to death and the use of MIA in six sites from five countries in sub-Saharan Africa and Asia [21] under the "Cause of Death through Minimally Invasive Autopsy (CaDMIA)" project [22]. Understanding the local culture appears extremely relevant for tailoring activities on this innovative and unfamiliar method for CoD investigation. In western Kenya this study examined cultural, social and religious norms around death, evaluated the willingness of next-of-kin and practitioners to know the CoD and its implications in different contexts, and explored community (including next-of-kin) attitudes and behaviours towards MIAs, in order to identify factors which could influence acceptance or refusal to allow MIA.

## Methods

### Study site and population

The study was conducted in Siaya County which is located in western Kenya near Lake Victoria. Within the county, a rural area of ~700km$^2$ is covered by a health and demographic surveillance system (HDSS) operated by the Kenya Medical Research Institute (KEMRI) where population-based reporting of births, deaths, and migrations is conducted longitudinally among ~240,000 people in ~70,000 households [23]. Households are visited twice annually to update demographic data, and a network of "village reporters" (akin to community health workers) provide information on deaths following their occurrence in the community. Verbal

autopsies are conducted on all reported deaths to identify common CoD using the WHO InterVA approach (http://www.interva.net/), and to facilitate evaluation of the changing patterns in these causes over the past decade [24–31]. The population is almost entirely of the Luo ethnic group, a community that practices polygamy, the majority of whom are subsistence farmers and small traders [32]. Siaya County is endemic for malaria and has an estimated HIV prevalence of 21% among adults, roughly triple the national prevalence. [33, 34]. The mortality rate for children under the age of five years in Siaya County was 167 per 1000 live births, and infant mortality rate was 112 per 1000 live births, as assessed in the 2011 Multiple Indicator Cluster Survey [34]. The study area is rural, and while served by over 30 government and faith-based health centres and dispensaries and a county referral hospital [23], there is nonetheless limited advanced medical treatment accessible to the population. Many health facilities have minimal diagnostic capability, limited cadres of staff, stock outs of available essential medicines and regular interruptions in electricity and other services. Transport of sick patients is generally conducted using private communal vans (*matatu*) on major inter-town routes, or motorcycle (*boda boda*), bicycle, or foot on local routes—exacerbating the difficulty of transporting seriously ill persons. Presently, the nearest facility where a CDA can be conducted is in Kisumu City (roughly one hour from the centre of Siaya county), thus the procedure is rare, unfamiliar to most residents and generally only known to be conducted in "police cases" (i.e., forensic investigations per legal requirements when foul play is suspected.) A study employing CDA was conducted in the region previously; while enrollment was limited many people in the region became aware of the study through stories and study sensitization efforts.

Culturally, death is a significant event in Luo communities, with large and expensive funerals held for adults to show respect to the deceased [35, 36]. Funerals may be held one week or more after death in order to allow family and other dignitaries to gather. Funerals for children, however, usually happen in the first 24–48 hours after death, and are much smaller events involving close family. Roughly 80% of all deaths occur at home in the Siaya study area (D. Obor, personal communication).

## Target study population

The target population for this study were community members who could best describe attitudes and practices related to death, both in personal and community terms. These included those affected directly by a recent death (e.g., next-of-kin to a deceased), persons within the health systems and program/policy makers in the health arena. Additionally, they were chosen to cover diverse groups within the population and a variety of service providers. For the purpose of this study respondents were divided into three groups according to their specific role in the context of deaths:

1. Next-of-kin: Persons residing in the KEMRI HDSS who experienced a death within their family during the study period. Next-of-kin were defined as the closest person to the deceased, usually though not necessary legally related to the deceased, who had decision making power on family health and death issues. This group of participants was divided into three sub-categories: (i) those who had experienced a very recent death of a family member (within 24 hours of the interview), (ii) those who has experienced a death in the preceding 1–7 days before the interview and (iii) those who had experience of a death of a family member between 30–40 days before the interview.

2. Community key informants: Persons knowledgeable about and/or influential in the community, such as local community and religious leaders, teachers, legal experts, advisory boards or local health or other sector committees, policy makers and governmental authorities;

3. Specialised key informants: Persons with specialised knowledge of end-of-life events, and the rituals and ethnic and religious norms and requirements for death-related events. These included funeral home personnel, religious leaders, community elders, traditional healers and birth attendants, medical facility personnel, community health personnel and VA interviewers.

## Study procedures

**Study design.** Kenya was one of five countries conducting CaDMIA, working from a similar protocol to assess the feasibility and acceptability of the MIA approach. (14) We adapted the common protocol to local circumstances. Sensitization meetings were held with Community Advisory Boards, local chiefs, politicians and village reporters to inform about the study aims and objectives. After sensitization, data collection through interviews, focus groups and observation proceeded, as described below:

**Data collection techniques:**

- To conduct interviews with next-of-kin immediately after death, the study coordinator received notification via mobile text message or call from village reporters who learned of the death from within the community. The study team visited to request consent and conduct a brief (typically under thirty minutes) semi-structured interview.

- Additional next-of-kin in-depth interviews were also conducted with different households approximately one month after burial; visits to consent and interview were coordinated with the HDSS VA team. These interviews lasted between thirty minutes and one hour, depending on the willingness of the respondent to discuss the issues.

- Community key informants were identified through suggestions from village reporters, local persons at community meetings, and from snowball sampling from other respondents. These key informant interviews typically lasted from one to one-and-half hours.

  Specialised key informants, such as facility- and community-based health care workers (medical doctors, clinical officers, nurses, lab technologists, community health workers and peer educators) and people whose work deals with death (pathologists, mortuary technicians, embalmers), were identified via suggestion from community members, community health workers and village reporters. We also made a purposive selection of staff at public and private health care facilities at various levels, based on specialist expertise, as well as workers at morgues and funeral homes serving the community. These key informant interviews typically lasted 30 minutes to longer than one hour, depending on the individual.

- Focus group discussions (FGDs) were held with several existing groups, including civic leaders and healthcare workers. Interviews as part of these groups lasted approximately 1–1 ½ hours.

- Study staff also attended events in the study community in order to observe rituals and interactions. Observations were made at home (home embalmment, night vigil, arrival of body from mortuary, or burial) or the mortuary (embalmment, viewing or removal for burial) or the church (for funeral prayers). Permission before the observation was sought from the event leader(s).

- Additionally, field notes were collected serendipitously, for example capturing comments made while waiting for a primary interview subject to be ready or when explaining the study to other people in the community. No identifying data were collected with any of these notes.

Interviews were conducted by either a single qualitative researcher or a two-person team from KEMRI, with the majority of the interviews conducted in Dholuo, the local language, or less frequently in Kiswahili or English. Interviews and FGDs followed an established guide which covered the topics of what happens around death in the community (e.g., procedures, practices and timing), perceived value and importance of knowing cause of death, and the respondent's concerns about and perceived benefits of MIA procedures. The guides offered open-ended questions, but research staff were also able to follow lines of inquiry that were raised during in-depth interviews using probing and follow-up questions. "MIA" was explained briefly to participants as a procedure in which needles would be used to take samples from the body of a deceased, with samples examined and tested in a laboratory to help understand what may have been the person's cause of death. Next-of-kin, specifically, were also asked whether hypothetically they would have permitted a MIA to have been performed on the recently deceased, why or why not, and under what circumstances they might have agreed. Observations were conducted by the qualitative research team witnessing body embalmment, night vigils, funerals, religious services and VA home visits.

### Data management and analysis

Audio contents of in-depth interviews, semi-structured interviews and FGDs were digitally recorded, and subsequently transcribed in the language of the interview, then translated to English by bilingual researchers. In the event where recording was unsuccessful or not permitted, field notes were used. Observations reports were recorded as field notes by staff attending community events.

Transcripts, observation reports and field notes (including from short/informal conversations) were coded by the Kisumu-based research team using Grounded Theory analysis. An initial generic outline of nodes and codes, based on the interview guides, was developed with other CADMIA sites to facilitate later cross-site analyses. Kenya-based staff worked collaboratively to develop a fuller coding frame which captured emerging themes responsive to the data from the Kenya site.

Data management and analysis were conducted using NVivo version 10 (QSR International Ltd, Australia, www.qsrinternational.com), software that facilitates the management, coding and keeps track of the analysis process of large sets of qualitative data.

### Ethical considerations

The study was approved by the local and national KEMRI ethical committee (# 2710), and the US Centers for Disease Control and Prevention relied upon KEMRI's review. Individual informed consent was read out to participants, and each received a copy of the study information sheet and the signed informed consent document. Audio-recordings and documents with identification data (e.g. names, directions to a household) were only accessible to research staff dealing directly with participants; all other data were stripped of identifiers during analysis.

### Results

One-hundred and twenty-nine (129) interviews were conducted for this study; 4 (3%) with next-of-kin within 24 hours after death, 17 (13%) with next-of-kin soon (within 1-7days) after death, 33 (26%) with next-of-kin after the burial of the deceased (within 30–40 days after death), 30 (23%) community key informants and 45 (35%) specialised key informants. Additionally, we held six FGDs; two with community elders and leaders, one with health care workers and three with village reporters (totaling 66 FGD participants). Five participant

observations were conducted, of body embalmment, removal of body from mortuary and arrival at home, night vigil and funeral service.

The demographic characteristics of the interviewees are described in a network-wide manuscript [14]. In brief, nearly all participants were of the Luo tribe, we had a slightly higher number of male than female respondents, and while most respondents were aged 30–49, about one-fifth were under 30 years of age and approximately one-third were aged 50 years or older. All but three respondents were Christians. While 10% of our respondents had no formal schooling, most respondents had attended primary or secondary school, and due to our inclusion of policy-makers and health professionals, approximately one third had professional or advanced degrees.

In our analysis, we did not find discernable differences in the perspectives that were offered among the three groups of next of kin; that is those interviewed within 24 hours, within one week or within 30–40 days after death. Therefore, we have categorized findings according to two major conceptual frameworks which took full shape as concrete themes emerged from the data: (1) factors that can positively affect acceptance of MIA, and (2) factors that can lead to rejection of MIA. The major nodes used in analysis are described in S1 Appendix.

## Conceptual framework for acceptance of MIA

Our research suggests a conceptual framework under which MIA and learning the CoD of a next-of-kin can be accepted (Fig 1).

**Desire to know CoD.**   Respondents described several situations in which there is an existing desire to know the CoD. This was discussed especially in cases of child and maternal death as well as in sudden death, when it was described that the family had a heightened interest to understand the cause. Respondents mentioned that when death occurs at a health facility there may be concerns and suspicions about treatment received, and therefore a desire to know the CoD to assess whether there was fault on the part of health care providers. Another example of the desire to know the CoD was when a death was suspected to be due to witchcraft or *chiraa* (contravening of tradition) and therefore considered to be "unnatural." Death due to witchcraft has implications for the family and for the memory of the deceased and therefore, can be a cause of consternation. Community members are eager to hear the medical CoD, creating an incentive for the family to have this information.

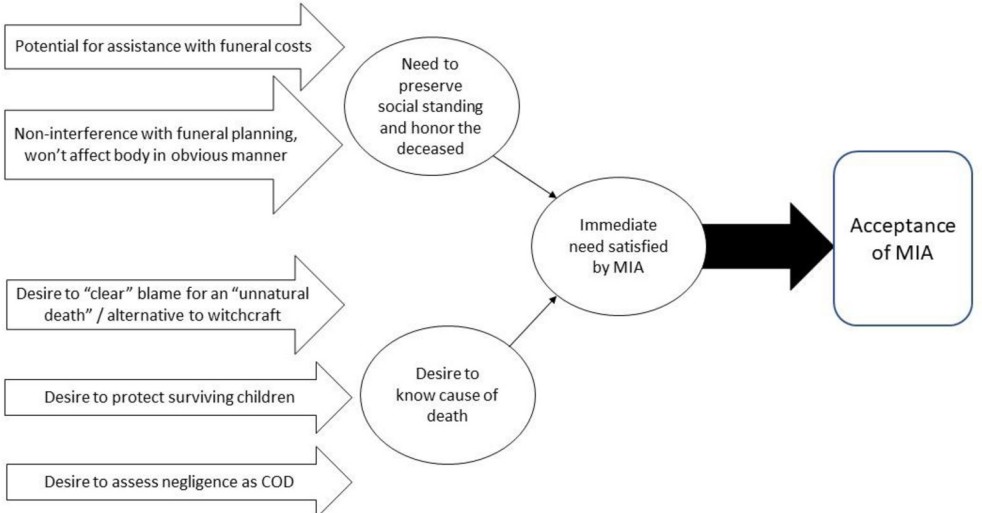

**Fig 1. Conceptual framework for acceptance of MIA.**

"*It is good in cases where there is a conflict or maybe someone thinks there was something [e. g., witchcraft] was done or you feel this is actually what happens . . . then it sort of resolves the conflict*"

{*Health care provider/Nurse, female*}

"*In my opinion I would have agreed for it [MIA] to be done, because most of the time when someone dies according to our belief someone had a hand in killing him or her. But instead the person died of sickness, so when MIA is done it will establish the cause of death, and this will be good. In our community someone doesn't just die without someone being pin-pointed as the killer. So I support MIA.*"

{*Next-of-kin, Elderly male*}

Parents spoke of a desire to protect their surviving children and other family members by having more knowledge about the cause of the child's death, and possibly being able to respond better if such illness were to occur in the future to a sibling or other child. Additionally, it was explained that mothers may be blamed for the death of a child, either for poor parenting or by causation an "unnatural" death due to *chiraa* or witchcraft (when someone else may cause the death of the child due to ill wishes). Some female respondents saw the possibility that MIA could possibly assist a mother who wants to clear suspicion that she caused the death, through either poor caretaking or due to *chiraa*. It was also expressed that a mother's own desire to know if she could have prevented the death would be a factor for acceptance.

"*. . .like babies, whereby they just die, this baby could not say what he or she was feeling and the mother is asking herself whether she made the mistake or who made the mistake. Was it the health care worker? Or what was not done? I think such questions will be answered.*"

{*Community leader, male*}

"*He was examined and nothing could be found ailing the baby, [he] could not be diagnosed, so I do not know very well what caused his death. The grandmother was saying maybe it is chiraa, that is why [we] used herbal medicines to treat the child too [long]. . .*"

{*Next-of-kin/mother of deceased child*}

We interviewed family members in ten cases of maternal death. Respondents spoke of the fact that in cases of maternal death the family may believe that the death occurred due to medical negligence or due to witchcraft, for example if it is believed that someone in the community or a co-wife wished ill to the mother. It was seen that MIA could provide some clarity in these situations, providing the husband and children with answers regarding the mother's death. It was also mentioned by several participants that the idea of postmortem procedures (more invasive than MIA) was already familiar in cases of maternal death as per traditional burial requirements it was necessary to separate a mother's and a fetus's body:

"*Yes [people can agree to MIA in maternal death cases] because those are simple procedures, you don't open up the body you only injure. . . because even within the villages when . . . an expectant mother passes on, there is a belief that the mother is not buried with the fetus, so what normally happens there is someone who is brought to remove the fetus from the mothers body and that is done at home so why not just injecting and taking some specimen.*"

{*Male key informant/ local public health researcher*}

**Knowing CoD offers future benefit to the community.** A small number of respondents, primarily health care providers and health administrators, described their belief that learning more about CoD could assist in improving medical care and public health for the community as a whole.

> *"They will see the benefit, because if they help someone who is still alive who was suffering from that illness, you must see the benefit."*

> {*Traditional birth attendant, female*}

> *"Knowing cause of death has one benefit, because it will make me take precaution because I'm still alive, it will make me take precaution maybe from what was happening that caused the death"*

> {*Male verbal autopsy interviewer, FGD participant*}

> *". . .it is of most benefit because sometimes we are unable to pin point what actually led to the death of someone, some patients die before we are able to make a definite diagnosis, we are working with a provisional diagnosis, we are yet to do some investigations, we are unable to do some investigations and in those cases, we can't completely say we know what caused the death because of limitation of tests and limitations of specialization amongst us"*

> {*Healthcare worker/clinician, male*}

> *"It is beneficial to the health professionals; it can help in the continuity of management of patient. May be the cause was hidden, so in future if you notice the same symptoms on a patient it can help you the direction to take"*

> {*Healthcare worker/nurse, female*}

Some respondents mentioned that the results of MIA should be put to immediate use, including possibly providing treatment to family of the deceased if a specific risk was identified during MIA.

> *"Maybe support. . . it can be in terms of medicine to protect the remaining individuals, so maybe the death of so-and-so made the community to be given such drug to protect the community."*

> {*Next-of-kin, male*}

**Serving the immediate needs of the family.** There is a history in the community of participants receiving small financial or in-kind incentives for participation in health studies, though the HDSS does not offer an incentive for participation in census activities. Many CaD-MIA respondents indicated that they were aware of the previous study in the region which had conducted CDA, and which offered the family 5000 Kenyan shillings (approximately 50 USD) to assist with the purchase of a coffin for the deceased. Respondents saw agreeing to participate in MIA as offering the potential to receive some type of incentive, which would help satisfy the pressing financial and logistical needs which arise at the time of death. Furthermore, there is a tradition throughout much of Kenya where friends and relatives of the family of the deceased make contribution toward funeral expenses. Referred to as "harambee" this tradition is used in a variety of other situations of urgent need as well, such as medical expenses and school fees. It was explained that Luo families face acute financial needs upon death, especially with the death of a senior family member, due to community expectations of an elaborate funeral to

honor the deceased. In our study communities, funerals for adults typically occur on the weekend following death, or later if critical family members are still awaited for funeral events or some matters related to the deceased had yet to be settled. It was explained that this latter situation may arise, in particular, in the case of agreeing on how to settle the estate of a man with more than one wife.

Many respondents described assistance with coffin purchase or other financial assistance by the program conducting MIA as an important motivation for agreeing to participate. In addition, respondents explained that if the MIA process keeps a body of an adult in the mortuary for several days and provides embalming after the MIA procedure, this was seen as an added value of participation, as these actions satisfy existing needs (waiting for the funeral) and reduce the financial burden on the family.

*"You know, nowadays, should someone face a problem and then receive assistance, the community will accept the study because people depend on assistance, and this is because of the challenges that people go through and the poor economic status."*

{*Next-of-kin, male*}

*". . .if you just go with nothing for the bereaved family, there are people who can say no to MIA."*

{*Key informant, male*}

**Minimal interference with the body.**   After learning about how MIA is conducted, respondents expressed that MIA would be accepted due the minimal effect on the body which they perceived as giving respect to the deceased. For some respondents MIA was contrasted to what they understood or had heard of previous complete autopsies conducted through another study—however few had personal experience of any postmortem procedure.

*"I would go for MIA due to reasons one, there is that maximum respect to the body, maybe you have seen how postmortem is done, the process of opening that body, cutting it is an inhumane way, I think MIA is more efficient than the other one, because you can cut and chop most precious part of the body, but for MIA that small portion is enough for the entire process, just a tissue, that is enough for the [minimally invasive] autopsy."*

{*Pharmacist, male*}

## Conceptual framework of factors that would hinder acceptance of MIA

Responses were also used to construct a conceptual framework of issues that present potential hindrances or barriers to conducting MIA (Fig 2). Broadly, these concerns were in place when learning CoD failed to serve any purpose identified by the family, or when conduct of MIA would exacerbate an already difficult period for the family.

**Limited value or use to know the CoD.**   Some respondents expressed the perception that learning the CoD would not serve a purpose for the family and therefore MIA did not serve a role. One reason given was the idea that death was "in the hands of God" and not under the control of humans, therefore making post-mortem investigations not useful. Others expressed their concern that MIA would not benefit the deceased and therefore did not serve a purpose.

*"For one, death is irreversible, once death has occurred, would that postmortem bring back my loved one? But when one is gone, in my case he or she is already gone."*

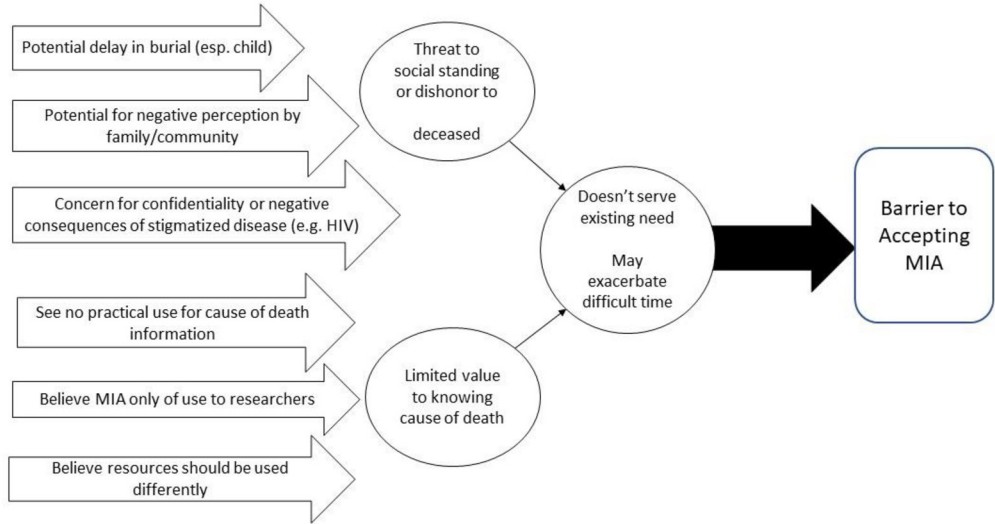

**Fig 2. Conceptual framework for barriers to MIA.**

{*Healthcare worker/nurse, female*}

"*It is very rare [to learn the cause of death] because it has never happened with us to have the cause of death because we say we are all from God and we shall all return to God, we do not go deep to know what really killed the person, we leave it to God. He brings and takes away and no questioning of that.*"

{*Muslim community leader, male*}

**Complications to learning the CoD.** Concerns about learning about an HIV diagnosis as part of learning the CoD were mentioned frequently, bearing in mind that Siaya County has one of the highest rates of HIV infection in Kenya. Some respondents mentioned an emphasis on and concern about confidentiality of the results. It was described that while it is well known that there is HIV in the Luo community, there was nonetheless concern about having others in the community learn about HIV in one's family.

"*. . . you know we still love our dead, and in a case where we have known, like in my case I know my husband has died of HIV, and now you are coming up with a tool that is going to determine and everybody is going to know what killed my husband, so actually it will be received with mixed feelings*"

{*Next-of-kin and healthcare worker, female*)

In the Luo community there is a tradition of widow-inheritance {Geissler, 2010 #34}, in which a brother or other male relative of the deceased may take on responsibility for a widow, one aspect of which is that they will engage in sexual relations. While somewhat diminishing, this practice is still in place especially in rural areas in part because women may have limited options for employment. Several respondents mentioned that findings about the husband's CoD, particularly in the case of HIV, could have implications for his surviving widow. It was explained that she can be stigmatized or find it difficult to get an "inheritor," and therefore may have difficulty to support herself and her children.

"... *now there is this issue of inheritance, so somebody would like to know, if the deceased was a man and somebody would like to inherit the wife, they would really want to know the cause of death*"

{*Key Informant, male*}

"*There is stigma in the community . . . Because of stigma, the community would not want to know the cause of death, including the immediate family because if people know so-and-so died of HIV the widow might not be inherited which is a practice that they still do. No one will inherit her because she is [HIV] positive.*"

{*Key informant, male*}.

**Prioritization of resources.**   Multiple respondents expressed concern that MIA would utilize resources which they felt would be better spent on "the living." Respondents were concerned that the use of transport, medical equipment and trained personnel should be devoted to saving lives rather than assessing the CoD for those who were already deceased. Community respondents expressed their experience that autopsies are costly and inaccessible, and concerns about the costs of conducting post-mortem studies (including MIA) were also raised by community leaders and health officials. Some respondents shared their opinion that conducting MIA would only be of benefit to researchers, rather than to families themselves.

"*That is what I'm trying to say. . . please try to find the illness when someone is still alive, stop looking for this thing [cause of death] when someone is dead. Please help people when they are still alive, my people.*"

{*Traditional Birth Attendant, female*}

"*She [the deceased] was bitten by a snake and CDC [CDC and KEMRI researchers working in the HDSS] did nothing about it. She was buying drugs for herself and sometimes she did not afford the drugs. I'm just wondering why the CDC could not help while she was still alive, until she is now dead is when the CDC comes to ask questions about her death.*"

(*Next-of-kin, male*)

**Negative impact on the family.**   In some cases, respondents shared concerns that agreeing to MIA could contribute to conflict within the family, either due to agreeing to the procedures at all or because acceptance of money from the MIA team (e.g., funds toward a coffin) could be viewed as "selling the body". It was also suggested that there could be resentment or jealousy toward the person(s) who would receive any funds that were provided.

"*Even in my case there were concerns. It is my firm nature which helped me out when people were now asking me how dare I give my mother's body to the CDC [researchers] who are going to take some body parts to sell and only pleasing me with 5000 shillings. I became so wild and told them the CDC did not take my mother's body because I did not have money*".

{*Next-of-kin, male, referring to participation in a previous postmortem study in the region*}

Another setting in which participation in MIA was seen as problematic for the family was in the case of death of a child or a stillbirth. Respondents described that burial procedures for children in Luo communities vary significantly from adults; typically the child will be buried by sundown on the day of death or the following day. Generally, only immediate family

members attend the burial, as a child is not yet regarded as a full member of the community. For this reason, there was some concern that MIA procedures could interfere with the normal timing of burial which some respondents saw as a barrier. It was also acknowledged the logistics of reaching the family in time to get consent for MIA when death had occurred at home may be more difficult in the case of children as compared to adults. This was particularly noted in the case of a stillborn child, as the deceased child would be taken home and buried very rapidly. It was described that this was seen as a means to spare the mother the emotional toll of having the deceased infant present.

> "*I think it is not easy [for MIA] to be carried [out] on stillbirth. This is because as Luos stillbirths are buried immediately. . . .Unless it happens in the hospital. Suppose my daughter had a stillbirth in the morning, by around this time [1100hrs], the body [of the child] would have been buried.*"

> {*Key Informant, female*}

It was infrequent that respondents spoke about a specific concern that participating in MIA would increase the emotional toll of experiencing a death, however one respondent expressed how adding an additional procedure, especially one done in the morgue, would have affected him:

> "*I did not want my child to first go to the mortuary. Secondly, to stay long so that the body will turn, I wanted . . . my daughter to maintain her freshness. I did not want anybody to touch her [at the mortuary].*"

> {*Next-of-kin, father of a deceased child*}

Also with regard to the death of a child, another respondent indicated that it would be difficult to find him in a state of mind to speak with anyone about allowing such a procedure:

> "*I don't know, but I don't think if you came to ask me to do such a procedure on my child in about two or three hours [after death], I would not have anything to do with you.*"

> {*Next-of-kin, male*}

Some respondents reported that MIA would not be accepted because they saw it as a form of body mutilation and they felt that there were cultural rules against this. This was, however, a theme that arose in relatively few settings.

> "*That procedure is actually very much condemned, because if somebody goes on puncturing the body of the deceased it is not actually good, if the patient was still in the ward the body can be punctured to find out the illness, but somebody just putting a knife on the body, that one no*! *It is condemned.*"

> {*Surgical theatre technician, male*}

**Religious concerns among Muslim respondents.** Muslims are a relatively small proportion of the population in western Kenya. We had the opportunity to speak with two Muslim respondents—a community leader and a clinical officer, who expressed some additional concerns with the conduct of MIA. They explained that taking the body of a deceased Muslim to the morgue could not be permitted, and there are specific times prescribed for burial, therefore

introducing concerns with potential delay in burial. Both nonetheless indicated the possibility of allowing MIA through involvement of Muslim organization leaders and if Muslims' concerns were addressed. It was emphasized that leaders in the community would need to approve of any procedures verses only getting individual approval.

"*We look at the Islam perspective; after looking the Islam perspective then after that there is a discussion and then they will come to resolve, okay, let us try it. But the consent has to depend on them [community leaders].*"

{*Muslim community leader, male*}

"*We have Muslim organizations who give directions to the Muslims, so even in those organizations we have officials, . . ., so through these organizations where their leaders are you can talk to them then they can tell you yes or no, because if you go down to the community level, that will be at an individual level.*"

{*Clinical officer who is Muslim, male*}

## Discussion

Our intensive qualitative research in Siaya County, Kenya suggests that the use of MIA and similar procedures to learn the CoD in our setting can be accepted if the implementation is effected in a manner that will address existing family and community needs and concerns, and will minimize the negative impact on the family. Death is a key event in Luo community life, with an imperative to honor and respect the deceased and to be seen by other community members as acting within expected norms. For the majority of respondents, MIA is not intrinsically unacceptable, rather participants needed to see a practical utility in order to consider participation.

The issue of practical matters stood out for these Kenyan participants. This is not unlike the findings by Cox *et al* who found that desire not to delay burial was the most common reason for declining an autopsy in Uganda [37]. Aside from being driven by practical and material needs at the time of death, a contribution to the family is in keeping with the widespread Kenyan tradition of "*harambee*" in which many community members, even those peripheral to the family such as workmates, make donations to assist with funeral costs. A financial contribution has a recognized cultural context in showing respect to the dead and to the family members. By coming in close contact with the family at the time of death, the research team has now become "involved," and thus the contribution is seen as both expected and appropriate in this setting. The expectation of respondents to receive a motivation/incentive to participate in MIA was clear, not only in terms of using funds to help with funeral expenses, but also by serving the existing needs to provide body preservation, storage and returning the body to a specified location for funeral proceedings.

Nonetheless, it is important that the relationship with the program conducting MIA not become coercive since, as stated by one respondent *"someone who has experienced loss cannot refuse help."* Because families are so driven by the practical needs underlying an appropriate funeral/burial, it seemed quite clear that to approach the family about participation without offering an appropriate contribution—on the grounds of learning cause of death or more esoteric service to the wider community—might simply be too far outside the realm of the family's frame of reference at the time of death, and thus would fail to address any of their needs or to provide the rest of the community for a frame of reference for participation. Finding the

proper point at which the contribution is a sign of respect and helpful to the family but not a cause of coercion or tension will be key to future postmortem work. Qualitative research should target this particular question.

We also found that there are some situations in which there is an existing interest to know the CoD of a loved one, especially in situations of child and maternal deaths and when a medical CoD may help to diffuse suspicions of witchcraft or poor medical treatment. The benefits of MIA for the family and the community will need to be conveyed clearly. It may be possible that intrinsic demand for MIA may increase as it becomes more familiar to the family and community members,. At the time of the study, the utility of CoD information was understood on a theoretical level, however conducting MIA was also a challenge to many respondents' sense of "just use" of resources. Some respondents associated any post-mortem study as part of academic or donor-driven schemes that would not be inherently useful for the community, rather expressing a prioritization of services for the living. This suggests the need to clearly connect any post-mortem investigations and the institution conducting them to visible health initiatives that benefit the living, representing the continuum of public health studies and actions. While respondents in Gurley *et al* may have been willing to see how MIA was of benefit to the community—and even that community leaders should have a role in decision-making [38]–our respondents in Kenya were somewhat more reserved (although it is difficult to compare an outbreak situation to more typical and ongoing health challenges). Some respondents in our study did make the connection to health care among survivors, by emphasizing that MIA findings should be used to direct and provide treatment, as might be relevant in the case of infectious disease or genetic illness that could affect siblings. It will be important to consider how to institute the use of MIA as truly a *surveillance* method with local utility and visible turn-around of information *versus* a "study" with more academic outcomes.

In conclusion, there was a willingness to accept MIA among our study participants, however it is also true that the community did not perceive a "need" for MIA. It may also be that the procedures were as yet so unfamiliar that the option of knowing CoD is not yet sufficiently recognized and therefore the desire to know CoD may increase over time. Certainly among healthcare workers MIA can help address a desire that already exist to improve their practice. Although we captured a heightened interest to know CoD in certain circumstances, the community's "intrinsic" desire to know CoD is presently limited. The initiation of routine MIA may therefore be most successfully accepted when it is introduced under a framework where the needs of next-of-kin are addressed by MIA including when ethically and socially appropriate reasons for participating in MIA have been established, and negative pressures reduced.

## Supporting information

**S1 Appendix. Nodes used in analysis.**
(DOCX)

## Acknowledgments

We would like to acknowledge the many families who allowed us to speak with them during a difficult time in their lives, and Dr. Emily Zielinski-Gutierrez who assisted with analysis and conceptualization of these findings.

The findings and conclusions in this paper are those of the authors and do not necessarily represent the official position of the U.S. Centers for Disease Control and Prevention.

## Author Contributions

**Conceptualization:** Kelvin Oruko, Maria Maixenchs, Penelope Phillips-Howard, Ariadna Sanz, Jaume Ordi, Clara Menéndez, Quique Bassat, Frank O. Odhiambo, Khatia Munguambe.

**Data curation:** Kelvin Oruko, Maria Maixenchs, Penelope Phillips-Howard, Maureen Ondire, Clarah Akelo.

**Formal analysis:** Kelvin Oruko, Maria Maixenchs, Maureen Ondire, Clarah Akelo.

**Funding acquisition:** Maria Maixenchs, Jaume Ordi, Clara Menéndez, Quique Bassat, Frank O. Odhiambo.

**Investigation:** Kelvin Oruko, Maria Maixenchs, Penelope Phillips-Howard, Maureen Ondire, Clarah Akelo, Clara Menéndez, Quique Bassat, Frank O. Odhiambo, Khatia Munguambe.

**Methodology:** Kelvin Oruko, Maria Maixenchs, Penelope Phillips-Howard, Jaume Ordi, Clara Menéndez, Quique Bassat, Frank O. Odhiambo, Khatia Munguambe.

**Project administration:** Kelvin Oruko, Ariadna Sanz, Quique Bassat, Frank O. Odhiambo, Khatia Munguambe.

**Resources:** Kelvin Oruko, Maria Maixenchs, Penelope Phillips-Howard, Maureen Ondire, Clarah Akelo, Frank O. Odhiambo, Khatia Munguambe.

**Software:** Kelvin Oruko, Maria Maixenchs, Maureen Ondire, Clarah Akelo, Frank O. Odhiambo.

**Supervision:** Kelvin Oruko, Quique Bassat, Frank O. Odhiambo, Khatia Munguambe.

**Validation:** Kelvin Oruko, Maria Maixenchs, Khatia Munguambe.

**Visualization:** Kelvin Oruko, Maria Maixenchs, Quique Bassat, Khatia Munguambe.

**Writing – original draft:** Kelvin Oruko, Maureen Ondire, Clarah Akelo, Quique Bassat, Khatia Munguambe.

**Writing – review & editing:** Kelvin Oruko, Maria Maixenchs, Penelope Phillips-Howard, Maureen Ondire, Clarah Akelo, Ariadna Sanz, Jaume Ordi, Clara Menéndez, Quique Bassat, Frank O. Odhiambo, Khatia Munguambe.

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
