## [Decision Letter · Decision Letter 0]

14 Sep 2020

PONE-D-20-21534

Conceptual frameworks for understanding the acceptability and feasibility of the minimally invasive autopsy to determine cause of death: Findings from the CADMIA Study in western Kenya

PLOS ONE

Dear Dr. Oruko,

Thank you for submitting your manuscript to PLOS ONE. After careful consideration, we feel that it has merit but does not fully meet PLOS ONE’s publication criteria as it currently stands. Therefore, we invite you to submit a revised version of the manuscript that addresses the points raised during the review process.

Please, address the comments from the reviewers and submit a revised version for further consideration.

We look forward to receiving your revised manuscript.

Kind regards,

Associate Professor Dr Muhammad Aziz Rahman,

MBBS, MPH, CertGTC, GCHECTL, PhD

Academic Editor

PLOS ONE

Journal Requirements:

3. Please note that in order to use the direct billing option the corresponding author must be affiliated with the chosen institute. Please either amend your manuscript or remove this option (via Edit Submission).

4. We note that Figure 1 in your submission contain map images which may be copyrighted. All PLOS content is published under the Creative Commons Attribution License (CC BY 4.0), which means that the manuscript, images, and Supporting Information files will be freely available online, and any third party is permitted to access, download, copy, distribute, and use these materials in any way, even commercially, with proper attribution. For these reasons, we cannot publish previously copyrighted maps or satellite images created using proprietary data, such as Google software (Google Maps, Street View, and Earth). For more information, see our copyright guidelines: http://journals.plos.org/plosone/s/licenses-and-copyright.

4.1.    You may seek permission from the original copyright holder of Figure 1 to publish the content specifically under the CC BY 4.0 license. 

4.2.    If you are unable to obtain permission from the original copyright holder to publish these figures under the CC BY 4.0 license or if the copyright holder’s requirements are incompatible with the CC BY 4.0 license, please either i) remove the figure or ii) supply a replacement figure that complies with the CC BY 4.0 license. Please check copyright information on all replacement figures and update the figure caption with source information. If applicable, please specify in the figure caption text when a figure is similar but not identical to the original image and is therefore for illustrative purposes only.

Reviewers' comments:

Reviewer's Responses to Questions

**Comments to the Author**

1. Is the manuscript technically sound, and do the data support the conclusions?

Reviewer #1: Yes

Reviewer #2: Yes

2. Has the statistical analysis been performed appropriately and rigorously? 

Reviewer #1: N/A

Reviewer #2: I Don't Know

3. Have the authors made all data underlying the findings in their manuscript fully available?

Reviewer #1: Yes

Reviewer #2: Yes

4. Is the manuscript presented in an intelligible fashion and written in standard English?

Reviewer #1: Yes

Reviewer #2: Yes

5. Review Comments to the Author

Reviewer #1: Well written paper but there are areas which need clarification, refer to the in-text comments. I was particularly looking forward to whether there were some sort of comparative reports mentioned between the 3 groups of which one group was in the immediate phase of grieving of which again, there was a large sample variation which also needs explanation and how the authors will control bias on reported statements given the sample variation. This needs clarity.

Reviewer #2: 1.Please arrange all heading and description in a orderly manner. In some page the heading is there but description is in next page.

2. There are some typos in the document. Please correct those typos.

3. During formulating questionnaire and checklist of qualitative part did you go for back to back translation, which should be unique one.

6. PLOS authors have the option to publish the peer review history of their article (what does this mean?). If published, this will include your full peer review and any attached files.

Reviewer #1: No

Reviewer #2: **Yes: **Dr Sheikh Mohammad Mahbubus Sobhan

---

## [Author Response · Author response to Decision Letter 0]

28 Oct 2020

Kindly, find responses to specific reviewer and editor comments in the attached document 'Responses to reviewers'

---

## [Editor Report · Decision Letter 1]

5 Nov 2020

Conceptual frameworks for understanding the acceptability and feasibility of the minimally invasive autopsy to determine cause of death: Findings from the CADMIA Study in western Kenya

PONE-D-20-21534R1

Dear Dr. Oruko,

We’re pleased to inform you that your manuscript has been judged scientifically suitable for publication and will be formally accepted for publication once it meets all outstanding technical requirements.

Kind regards,

Associate Professor Dr. Muhammad Aziz Rahman,

MBBS, MPH, CertGTC, GCHECTL, PhD

Academic Editor

PLOS ONE

---

## [Editor Report · Acceptance letter]

3 Dec 2020

PONE-D-20-21534R1 

Conceptual frameworks for understanding the acceptability and feasibility of the minimally invasive autopsy to determine cause of death: Findings from the CADMIA Study in western Kenya 

Dear Dr. Oruko:

I'm pleased to inform you that your manuscript has been deemed suitable for publication in PLOS ONE. Congratulations! Your manuscript is now with our production department. 

Kind regards, 

on behalf of

Associate Professor Dr. Muhammad Aziz Rahman 

Academic Editor

PLOS ONE